# Synthesis, Structural and Physicochemical Characterization of a Titanium(IV) Compound with the Hydroxamate Ligand *N*,2-Dihydroxybenzamide

**DOI:** 10.3390/molecules26185588

**Published:** 2021-09-15

**Authors:** Stamatis S. Passadis, Sofia Hadjithoma, Panagiota Siafarika, Angelos G. Kalampounias, Anastasios D. Keramidas, Haralampos N. Miras, Themistoklis A. Kabanos

**Affiliations:** 1Section of Inorganic and Analytical Chemistry, Department of Chemistry, University of Ioannina, 45110 Ioannina, Greece; stamatispassadis@hotmail.com; 2Department of Chemistry, University of Cyprus, Nicosia 2109, Cyprus; hadjithoma.sofia@ucy.ac.cy; 3Physical Chemistry Laboratory, Department of Chemistry, University of Ioannina, 45110 Ioannina, Greece; p.siafarika@uoi.gr; 4Institute of Materials Science and Computing, University Research Center of Ioannina (URCI), 45110 Ioannina, Greece; 5School of Chemistry, University of Glasgow, Glasgow G12 8QQ, UK

**Keywords:** titanium(IV) oxo-clusters, band gap modification, multinuclear NMR, ESI-MS studies, photoluminescence

## Abstract

The siderophore organic ligand *N*,2-dihydroxybenzamide (H_2_dihybe) incorporates the hydroxamate group, in addition to the phenoxy group in the ortho-position and reveals a very rich coordination chemistry with potential applications in medicine, materials, and physical sciences. The reaction of H_2_dihybe with TiCl_4_ in methyl alcohol and KOH yielded the tetranuclear titanium oxo-cluster (TOC) [Ti^IV^_4_(μ-O)_2_(HOCH_3_)_4_(*μ*-Hdihybe)_4_(Hdihybe)_4_]Cl_4_∙10H_2_O∙12CH_3_OH (**1**). The titanium compound was characterized by single-crystal X-ray structure analysis, ESI-MS, ^13^C, and ^1^H NMR spectroscopy, solid-state and solution UV–Vis, IR vibrational, and luminescence spectroscopies and molecular orbital calculations. The inorganic core Ti_4_(*μ*-O)_2_ of **1** constitutes a rare structural motif for discrete Ti^IV^_4_ oxo-clusters. High-resolution ESI-MS studies of **1** in methyl alcohol revealed the presence of isotopic distribution patterns which can be attributed to the tetranuclear clusters containing the inorganic core {Ti_4_(*μ*-O)_2_}. Solid-state IR spectroscopy of **1** showed the presence of an intense band at ~800 cm^−1^ which is absent in the spectrum of the H_2_dihybe and was attributed to the high-energy ν(Ti_2_–*μ*-O) stretching mode. The *ν*(C=O) in **1** is red-shifted by ~10 cm^−1^, while the *ν*(N-O) is blue-shifted by ~20 cm^−1^ in comparison to H_2_dihybe. Density Functional Theory (DFT) calculations reveal that in the experimental and theoretically predicted IR absorbance spectra of the ligand and Ti-complex, the main bands observed in the experimental spectra are also present in the calculated spectra supporting the proposed structural model. ^1^H and ^13^C NMR solution (CD_3_OD) studies of **1** reveal that it retains its integrity in CD_3_OD. The observed NMR changes upon addition of base to a CD_3_OD solution of **1**, are due to an acid–base equilibrium and not a change in the Ti^IV^ coordination environment while the decrease in the complex’s lability is due to the improved electron-donating properties which arise from the ligand deprotonation. Luminescence spectroscopic studies of **1** in solution reveal a dual narrow luminescence at different excitation wavelengths. The TOC **1** exhibits a band-gap of 1.98 eV which renders it a promising candidate for photocatalytic investigations.

## 1. Introduction

The design, synthesis, and physicochemical characterization of polyoxo-titanium clusters (PTCs) have been an active research area over the last decade, due to their interesting electronic properties for various applications in nanotechnology [1,2,3,4,5,6], photocatalytic hydrogen production [7,8,9], degradation of environmental pollutants [10,11], catalysis [12,13,14,15,16,17], solar energy conversion [18,19], and copolymerization of carbon dioxide [20]. At this point, it is worth noting that according to an excellent review, published very recently, the chemistry of group IV elements is underexplored [21].

The knowledge of the structural features of PTCs from single-crystal X-ray structure analysis is of fundamental importance to predict the binding modes of various ligands to TiO_2_ since the PTCs are considered solution-processable molecular analogs of TiO_2_. Moreover, the 3.20 eV band-gap of TiO_2_ limits its applications in photocatalysis [22,23]. However, the use of strong organic chelators allows the modulation of the band-gap with subsequent Vis-NIR absorption by PTCs to appropriate values, which is a fundamentally important parameter for practical applications [24].

Siderophores are low molecular weight organic compounds that are produced by microorganisms and plants suffering from iron deficiency [25]. Siderophores have received much attention in recent years due to their potential applications in environmental research [26]. Siderophores are divided into three main families depending on the characteristic functional group, i.e., hydroxamates, catecholates, and carboxylates [27].

Metal-hydroxamate has greater resistance to hydrolysis [28] in comparison to carboxylic acids and greater electronic coupling over carboxylic and phosphonic acids that ensures efficient electron transport [29,30]. The main binding modes of hydroxamate with metal ions are shown in Appendix A [28].

The organic molecule *N*,2-dihydroxybenzamide (H_2_dihybe) (Figure 1A) incorporates a hydroxamate group in addition to the phenoxy group in the ortho-position and exhibits a rich coordination chemistry with many transition metals [31,32,33,34,35,36,37,38,39,40] with potential applications in various fields ranging from medicine [41,42,43] to materials [44], and physical sciences [40,45].

The ligand H_2_dihybe has been used previously in the synthesis of titanium(IV) oxo-clusters [47] under hydrothermal conditions to give three TOCs, namely: [Ti_6_(*μ*-O)(*μ*_3_-O)_2_(O*^i^*Pr)_10_(OOCCH_3_)_2_(dihybe)_2_]; [Ti_7_(*μ*_3_-O)_2_(OEt)_18_(dihybe)_2_]; and [Ti_12_(*μ*-O)_4_(*μ*_3_-O)_4_ (OEt)_20_(dihybe)_4_]. The ligand in these three TOCs interacts with the titanium(IV) in its iminol tri-deprotonated *μ*_3_-*η*^1^:*η*^2^:*η*^1^:*η*^1^ form (Figure 1B).

The formation of various metal–siderophore complexes (nuclearity-binding modes of siderophore) among other factors greatly depends on the pH of the reaction mixture since there is a competition between free protons and metal ions for the free siderophore ligands.

To further explore the coordination chemistry of *N*,2-dihydroxybenzamide (H_2_dihybe) (Figure 1) with titanium(IV), we explored the interaction of Ti^IV^Cl_4_ with H_2_dihybe at low pH under mild conditions (room temperature). Herein, we report the synthesis, structural, and physicochemical characterization of the tetranuclear TOC [Ti^IV^_4_(μ-O)_2_(HOCH_3_)_4_(*μ*-Hdihybe)_4_(Hdihybe)_4_]Cl_4_^.^10H_2_O^.^12CH_3_OH (**1**). The cluster **1** constitutes a rare example of a discrete TOC containing the inorganic core {Ti_4_(*μ*-O)_2_}. Spectroscopic studies in the solid state revealed a reduced bandgap value of 1.98 eV. In addition, this compound produced dual emission because of changes in the emitting light with variations in the excitation wavelength. The materials with dual emission properties such as **1** can be applied in OLED devices, allowing tuning of the color of the emitting light depending on the voltage applied to the device [47].

## 2. Results and Discussion

### 2.1. Synthesis of ***1*** and Comparison with the Reported Higher Nuclearity TOCs/H_2_dihybe 

The synthesis of the TOC **1** takes place according to equation 1 and the produced HCl is responsible for the low pH (1.5) of the system and presumably for the mono-deprotonation of the eight ligands of **1**.
4TiCl_4_ + 8H_2_dihybe + 8KOH → [Ti_4_(*μ*-O)_2_(Hdihybe)_8_]Cl_4_ + 8KCl + 4HCl + 6H_2_O(1)(1)

In Equation (1), the molar ratio (mr) of TiCl_4_/H_2_dihybe is 1:2, while in the reported [46] Ti^IV^/H_2_dihybe TOCs {[Ti_6_(*μ*-O)(*μ*_3_-O)_2_(O*^i^*Pr)_10_(OOCCH_3_)_2_(dihybe)_2_]; [Ti_7_(*μ*_3_-O)_2_(OEt)_18_(dihybe)_2_]; and [Ti_12_(*μ*-O)_4_(*μ*_3_-O)_4_(OEt)_20_(dihybe)_4_]} the mr of Ti(O*^i^*Pr)_4_/H_2_dihybe is four and this means that more positions of the coordination sphere of titanium(IV), in our case, are occupied by the donor atoms of the ligand which precludes the formation of bigger clusters. On the other hand, the use of Ti(O*^i^*Pr)_4_ with the very strong base ^−^O*^i^*Pr, [deprotonated HO*^i^*Pr (pKa = 16.5) is a stronger base than H_2_dihybe (pKa = 9.57), thus, in these solutions, the ligand was deprotonated, ^−^Hdihybe], the high temperature and high pressure under the hydrothermal conditions lead to the formation of the tri-deprotonated ligand which is capable of bridging more metals and thus, leading to the formation of higher nuclearity TOCs.

From all the above, it is clear that the low molar ratio of titanium(IV)/H_2_dihybe, low pH, and room temperature lead to the formation of low nuclearity TOCs. The ligation of the ligand to Ti^IV^ makes the hydroxamic proton [~C(O)NHO-**H**] more acidic, and thus, despite the low pH, results in the deprotonation of the ligand.

### 2.2. Description of the Structure

Interatomic distances and bond angles relevant to the Ti(**1**) coordination sphere are listed in Table 1. The molecular structure of the cation [Ti^IV^_4_(*μ*-O)_2_(HOCH_3_)_4_(*μ*-*η*^1^,*η*^2^-Hdihybe-*O*,*O′*)_4_(*η*^1^,*η*^1^-Hdihybe-*O*,*O*′)_4_]^4+^ of **1** is presented in Figure 1 which is composed of two dinuclear [Ti^IV^_2_(HOCH_3_)_2_(*μ*-*η*^1^,*η*^2^-Hdihybe-*O*,*O′*)_2_(*η*^1^,*η*^1^-Hdihybe-*O*,*O*′)_2_]^4+^ (Figure 2A) units interlinked through two *μ*-bridging oxygen atoms (Figure 2B). Each titanium(IV) atom in **1** is bonded to two mono-deprotonated Hdihybe^–^ligands, one of which acts as a bidentate-*O*,*O*′ chelate through the carbonyl and the deprotonated hydroxamate oxygen atoms (see Figure 2a) and the other one as a chelate-bridging-*O*,*O*′ through the same oxygen atoms (see Figure 2b). All the titanium centers in **1** are seven-coordinate, with an O_7_ donor set, in a pentagonal bipyramidal environment and are sharing one of their edge (Figure 2B) in the {Ti_2_} structural unit and through a corner in the Ti(1)-O(15)-Ti(1)‴ unit (Figure 2B). The Ti(1)∙∙∙Ti(1)’ and Ti(1)∙∙∙Ti(1)” distances within the two dimeric {Ti_2_} units are 3.515(1) and 3.561(1) Å, respectively, while the Ti(1)-O(15)-Ti(1)‴ angle is 165.8(1)°. The inorganic core {Ti_4_(*μ*-O)_2_} constitutes a rare example of such a structural motif for discrete {Ti^IV^_4_} oxo-clusters. The other two examples which have been reported are the following: (NH_4_)_6_[Ti^IV^_4_(*μ*-O)_2_(C_2_H_2_O_3_)_4_(C_2_H_3_O_3_)_2_(O_2_)_4_] [48,49] and [Ti^IV^_4_(*μ*-O)_2_(*µ*-OEt)_4_(κ^3^-tbop)_4_] [50].

### 2.3. ESI-MS Spectrometry

In an effort to further characterize the tetranuclear titanium(IV) cluster in solution, we employed high-resolution ESI-MS to unambiguously determine the structural integrity and composition of the titanium-based species in solution [51,52]. The low *m*/*z* region of the negative ion mass spectrum of **1** exhibits two characteristic sets of isotopic distribution patterns (Figure 3) which can be attributed to the dinuclear and trinuclear fragments of **1** and are centered at: ca. (a) 698.98, 736.93, and 768.95 *m*/*z* with the formulae of {Ti^III^_2_O_2_(OCH_3_)_3_(C_7_H_5_NO_3_)_3_(OH_2_)H_6_}^–^, {Ti^III^_2_O_2_(C_7_H_5_NO_3_)_4_H_5_}^–^ and {Ti^III^_2_O_2_(OCH_3_) (C_7_H_5_NO_3_)_4_H_6_}^–^ for the dinuclear fragment and at (b) 895.911 and 931.90 *m*/*z* with the formulae of {Ti^III^_3_O_2_(OCH_3_)(C_7_H_5_NO_3_)_4_(HOCH_3_)_2_(OH_2_)H_3_}^–^ and {Ti^III^_3_O_2_(OCH_3_)(C_7_H_5_NO_3_)_4_(HOCH_3_)_2_(OH_2_)_3_H_3_}^–^ for the trinuclear fragments, respectively. The observation of different oxidation states for the metal centers and the presence of other fragments during the studies is due to the ionization and transfer process and has been observed in numerous occasions [53,54,55,56,57].

The high *m*/*z* region of the negative ion mass spectrum of **1** (Figure 4) exhibits characteristic isotopic distribution patterns which can be attributed to the tetranuclear cluster containing the inorganic core {Ti_4_O_2_} and are centered in the region ca. 1078–1315 *m*/*z*. See Table 2 for the assigned species. The partial fragmentation during the ESI-MS studies provides additional information on the relevant stability of the fragments that can exist in solution. Thus, this can provide an indication of the potential assembly pathway followed during the formation of the tetranuclear species which can be formed by the combination of smaller dimeric fragments, e.g., 2 × {Ti_1_} → 2 × {Ti_2_} → {Ti_4_}.

### 2.4. IR Spectroscopy

Quantitative agreement between experimental and theoretical spectra predicted by ab initio DFT/B3LYP calculations with the Los Alamos National Laboratory 2 double zeta (LanL2DZ) split-valence basis set. The specific basis set is an ideal choice for quantum mechanical calculations for complexes whose centers are first-row transition metals, such as titanium. The calculations of such molecules with the LANL2DZ basis set and the fact that the complex has *D_2_* point group symmetry are characterized by a relatively short computational time. Considering the LANL2DZ basis set, a reason for the reduction in the computational time is that ECP (Effective Core Potential) plus double zeta on Na-Bi is used in this basis set. ECP describes the inner electron orbitals and so no basis functions are required for them. Another advantage of this basis set is that it includes relativistic effects, but does not include polarization functions. Τhese results add detailed confidence to our present understanding of the chemistry of the systems. The experimental spectra of the ligand and complex, denoted as H_2_dihybe and **1**, respectively, in the solid state at ambient conditions are given in Figure 5, allowing for direct comparison. At a first glance, we observe that the spectrum of the H_2_dihybe is less complicated compared to the spectrum of the complex **1**. Furthermore, its energy minimum is lower as expected for a simpler structure. In the high-frequency spectral region, only the N–H, OH, and C–H stretching modes are detected for both spectra and do not provide any significant information concerning complexation. Thus, we have chosen to focus our attention on the low-frequency region, the so-called fingerprint region. This part of the spectrum is more informative in relation to the comprehensive understanding of the structural features of the titanium complex. In this region, a larger number of bands are present. Besides the fact that these bands are sharper and of higher intensity, also evidenced is a strong band overlapping that implies the inherent structural complexity of the studied complex. An intense band at ~800 cm^−1^ is observed in the spectrum of the complex which is absent in the spectrum of the H_2_dihybe. This band is attributed to the titanium complex formation and more specifically is assigned to the high-energy ν(Ti_2_–*μ*-O) stretching modes. The low-frequency band of H_2_dihybe observed at ~780 is red-shifted to 770 cm^−1^ in the spectrum of complex, while the band of H_2_dihybe at ~744 cm^−1^ remains in the same frequency with much lower absorbance. Additional relative absorbance changes are observed between the two spectra. The formation of the complex is expected to affect the vibrational frequency and/or absorbance of the C=O, C-O, C-N, and N-O bonds. Indeed, the frequency of C=O is red-shifted ~10 cm^−1^. This band is observed at ~1577 cm^−1^ and at ~1567 cm^−1^ in the spectrum of H_2_dihybe and complex, respectively. The bands at ~1250 cm^−1^ and ~1360 cm^−1^, attributed to C-O and to C-N, respectively, exhibit only a significant absorbance decrease, while there is no frequency shift upon formation of the complex formation. On the contrary, the band assigned to the N-O vibration is blue-shifted from ~1050 cm^−1^ to ~1070 cm^−1^ without any additional absorbance variation after complex formation. The experimental and theoretically predicted by ab initio DFT/B3LYP/LANL2DZ IR absorbance spectra of the H2dihybe and Ti-complex **1** are presented in Appendix A, respectively. All the main bands detected in the experimental spectra are also present in the calculated spectra supporting our proposed structural model. Any differences observed in intensities and frequencies are reasonable, considering that the calculation was performed in the vapor state without the presence of any additional interactions. The IR absorbance data are indicative of a structural rearrangement and complex formation which further support the structural information revealed by the rest of the experimental techniques utilized in the present study.

### 2.5. NMR Spectroscopy 

The ^1^H and ^13^C NMR chemical shifts of the CD_3_OD solutions of the H_2_dihybe and **1** are collected in Table 3. The ^1^H NMR spectrum of a CD_3_OD solution of the H_2_dihybe (Figure 6A) gave two doublets of doublets at 7.669, 6.928 ppm and two triplets of doublets at 7.338, 6.897 ppm assigned to the protons attached to carbon atoms C(d), C(a), and C(b), C(c), respectively.

The ^1^H NMR spectrum of the CD_3_OD solution of **1** shows peaks of the same multiplicity with H_2_dihybe, however, the peaks were shifted to lower field suggesting ligation of the ligand to Ti^IV^; with Δ*δ* for a, b, c, d protons 0.10, 0.07, 0.11, 0.16 ppm, respectively (Figure 6B). The ^1^H NMR spectrum of **1** shows only one set of peaks for the complex. However, the crystal structure of **1** (Figure 6) shows two mono-deprotonated Hdihybe^−^ ligands bound to each titanium atom with two different modes of ligation (Figure 2), and this means that either the complex **1** in solution does not have the same structure as the solid state or there is a fast-chemical exchange between the bridged and the non-bridged Hdihybe^−^ ligands. ESI MS experiments indicate that the complex retains its solid-state structure in CD_3_OD, therefore, the bridged and the non-bridged Hdihybe^−^ ligands cannot be distinguished by ^1^H NMR because they exchange fast. In addition, the chemical shifts of the bridged and the non-bridged Hdihybe^−^ ligands are not expected to be very different, thus, any exchange will result in their coalescence. Chemical exchange between the two coordinated Hdihybe^−^ ligands is supported by the much broader ^1^H peaks of **1** than the respective peaks of H_2_dihybe. The carbonylic ^13^C NMR peak of the CD_3_OD solution of **1**, determined by 2D grHSQC and gr HMBC spectroscopies (Appendix A), shows the largest shift compared to the respective peak of the ligand, Δ*δ* = −3.9 ppm, suggesting coordination of Ti^IV^ to the carbonylic oxygen atom [24,58,59,60,61].

In addition to the peaks originated from **1**, the ^1^H NMR spectrum gave peaks originated from a minor species (10%) and the free ligand (12%) (Figure 6). The chemical shifts of ^1^H and ^13^C, as they have been found from 2D {^1^H} grCOSY (Appendix A) and 2D {^1^H, ^13^C) grHSQC (Appendix A), are the same as **1** except C(d)-**H** proton which is shifted to the lower field (0.110 ppm) than the respective proton of **1** (Table 3). Apparently, the minor species **2** and **1** have a similar structure. Looking at the structure of **1**, one can suggest various isomers. The most possible one is that with the non-bridged Hdihybe^−^ ligated to Ti^IV^ at different orientations (Figure 3).

The 2D {^1^H} NOESY-EXSY spectrum (Figure 7) of the CD_3_OD solution of **1** gave NOESY cross peaks between the neighboring aromatic protons and EXSY cross-peaks between **1** and the free ligand, the minor species and ligand as well as between **1** and the minor species. The intensity ratios of the cross-peaks vs the diagonal were similar for all exchange processes suggesting that the conversion of **1** to the other isomer (Figure 3) is intermolecular. 

In addition, the 2D {^1^H} NOESY-EXSY spectrum showed two NOESY cross-peaks between protons H(**b**) and H(**d**) (See Figure 6) having the same phase with the diagonal peaks (Figure 7, blue circles). Despite the opposite phase than the expected one, these peaks should be originated from NOESY interactions; the possibility of these peaks originating from a chemical exchange is improbable. The phase of this peak is attributed to the fast exchange between the bridged and non-bridged Hdihybe^−^. These NOESY signals are assigned to the interactions between the H(**b**) and H(**d**) protons of different non-bridged Hdihybe^−^ ligands, each of which belongs to one of the two parallel planes, defined from two Ti^IV^ and four Hdihybe^−^ ligands (Figure 4). Apparently, this supports the complex in retaining its tetranuclear structure in agreement with the ESI measurements.

The 2D {^1^H} NOESY-EXSY spectrum (Figure 8) of the CD_3_OD solution of **1** gave NOESY cross-peaks between the neighboring aromatic protons and EXSY cross-peaks between **1** and the free ligand. In addition, it showed and two NOESY cross-peaks between protons **d** and **b** (Figure 8, blue circles).

In contrast to the spectrum of **1** without a base, the peaks now have the expected negative phase, assigned to the significant decrease in the complex’s lability after the addition of the base to the solution. The NOESY interactions between protons **b** and **d** suggest that the complex retains its tetra-nuclear structure after the addition of the base, permitting interactions between the protons of the parallel aromatic rings (Figure 4). Another observation that supports a similar tetranuclear structure before and after the addition of the base, is the fact that the NMR spectra of **1** after the addition either of one or four equivalents per equivalent of **1** are the same (Appendix A). Thus, supporting that the NMR changes observed after the addition of the base in the CD_3_OD solution of **1**, is a result of acid–base equilibrium and not a change in the Ti^IV^ coordination environment, and the decrease in the complex’s lability is due to the better electron-donating properties of the ligand after its deprotonation as shown in Figure 5.

### 2.6. Solution UV–Vis and Luminescence Spectroscopies

Figure 9 shows the solution UV−Vis spectra of the MeOH solutions of compound **1** without and with the presence of a base (But_4_NOH), The MeOH solution of **1** gave two peaks at 310 nm (38 cm^−1^M^−1^) and 242 nm (71 cm^−1^M^−1^). The peaks are attributed to n-π* and π-π* electronic transitions. The peak at 310 nm is characteristic for phenolate groups. After the addition of But_4_NOH, the peaks remain the same, but their intensity increased significantly. The results support that the structure of **1** remains the same in solution after the addition of the base, whereas the increase in the intensity of the peak at 310 nm is assigned to deprotonation of the phenol, in line with the NMR experiment.

Figure 10 shows the luminescence spectra of the MeOH solutions of compound **1** in the presence of a base (But_4_NOH), The MeOH solution of **1** gave dual-luminescence, emitting light at 620 nm for excitation wavelength 524 nm and at 573 nm for excitation wavelength 490 nm. The emitting peaks are relatively sharp with linewidths ~30 nm. The intensity of the emitting light was doubled after the addition of an equimolar quantity of base in the MeOH solution of **1**, however, the excitation and emission wavelengths remain the same suggesting that the structure in solution remains the same after the addition of the base. The increase in the intensity is assigned to the phenol deprotonation.

### 2.7. Solid-State UV−Vis Spectroscopy

Figure 11 shows the solid-state UV−Vis spectra of the compound **1** and the ligand H_2_dihybe. The band gap for the compound **1** was found to be 1.98 eV and was calculated from the solid-state spectrum by the Kubelka−Munk method [62] (Appendix A). This low band gap value for compound **1** reveals its potential use as a semiconducting photocatalyst. 

## 3. Materials and Methods

### 3.1. Experimental Details

All chemicals and solvents were purchased from Sigma-Aldrich and Merck (Saint Louis, MO, USA), were of reagent grade, and were used without further purification, except TiCl_4_, which was distilled under high vacuum just prior to use. C, H, and N analyses were conducted by the microanalytical service of the School of Chemistry, the University of Glasgow. FT-IR transmission spectra of the compounds, in KBr pellets, were acquired using a Bruker Alpha spectrophotometer (Bruker, Billerica, MA, USA) in the 4000−400 cm^−1^ range. The UV−Vis diffuse reflectance spectra were recorded at room temperature on an Agilent Cary 60 UV−Vis spectrophotometer (Agilent Technologies, Santa Clara, CA, USA). The UV–Vis and the luminescence solution spectra were acquired on a Shimadzu UV-2600i UV–Vis Spectrophotometer (Shimadzu, Nagoya, Japan) and on a Jasco Spectrofluorometer FP-8300 (JASCO, Mary’s Court Easton, MD 21601, USA), respectively, at room temperature.

### 3.2. Synthesis of [Ti^IV^_4_(μ-O)_2_(HOCH_3_)_4_(μ-η^1^,η^2^-Hdihybe-O,O′)_4_(η^1^,η^1^-hdihybe-O,O′)_4_]Cl_4_^.^10H_2_O^.^12CH_3_OH *(**1**)*

To a stirred methyl alcohol solution (4 mL) were successively added *N*,2-dihydroxybenzamide (H_2_dihybe) (139.7 mg, 0.912 mmol) and TiCl_4_ (0.05 mL, 86.5 mg, 0.456 mmol). The colorless solution of the ligand turned orange upon the addition of TiCl_4_. Then, solid ΚOH (51.1 mg, 0.912 mmol) was added in one portion. The solution was filtered, and the orange filtrate (pH = 1.5) was kept at ≈4 °C for 9–10 days during which period 90.0 mg of orange crystals of compound **1** were formed. The crystals were filtered off and dried at an ambient atmosphere (≈20 °C). (Yield: 35%, based on TiCl_4_). Elemental anal. calc. for (C_72_H_132_N_8_O_52_Cl_4_Ti_4_, *M*_r_ = 2275.112 g mol^−1^): C, 38.01; H, 5.85; N, 4.92; found: C, 37.98; H, 5.81; N, 4.95.

### 3.3. X-ray Crystallographic Details

A suitable single crystal was selected and mounted onto a rubber loop using Fomblin oil. Single-crystal X-ray diffraction data of **1** was recorded on a Bruker Apex II Quazar CCD diffractometer (Bruker, Bremen, Germany) (λ (MoK*_α_*) = 0.71073 Å) at 150 K equipped with a graphite monochromator. Structure solution and refinement were carried out with SHELXS-97 **[63]** and SHELXL-97 **[64]** using the WinGX software package [65]. Data collection and reduction were performed using the Apex2 software package. Corrections for the incident and diffracted beam absorption effects were applied using empirical absorption corrections [66]. All the atoms and most of the carbon atoms were refined anisotropically. Solvent molecule sites were found and included in the refinement of the structures. Final unit cell data and refinement statistics for compounds **1** are collated in Table 4. The crystallographic data for compound **1** (CCDC **1**: 2096669) can be obtained free of charge from the Cambridge Crystallographic Data Centre, 12, Union Road, Cambridge, CB2 1EZ; fax:(+44)-1223-336-033, deposit@ccdc.cam.ac.uk. 

### 3.4. ESI MS Experimental Details 

All MS data were collected using a Bruker Q-trap, time-of-flight MS (Maxis Impact MS, Bremen, Germany) instrument supplied by Bruker Daltonics Ltd. The detector was a time-of-flight, micro-channel plate detector and all data was processed using the Bruker Daltonics Data Analysis 4.1 software, whilst simulated isotope patterns were investigated using Bruker Isotope Pattern software and Molecular Weight Calculator 6.45. The calibration solution used was the Agilent ES tuning mix solution, Recorder No. G2421A, enabling calibration between approximately 100 *m/z* and 3000 *m/z*. This solution was diluted 60:1 with MeCN. Samples were dissolved in MeOH and introduced into the MS via direct injection at 180 µL h^−1^. The ion polarity for all MS scans recorded was negative, at 180 °C, with the voltage of the capillary tip set at 4000 V, endplate offset at −500 V, funnel 1 RF at 300 Vpp, and funnel 2 RF at 400 Vpp. 

### 3.5. FT-IR Spectroscopy

The FT-IR spectra of the ligand and the complex were recorded in the 4000–400 cm^−1^ mid-infrared range on a Bruker Apha FT-IR spectrophotometer (Bruker, Billerica, MA, USA) with 256 scans at a resolution of 2 cm^−1^. All samples in the solid form were ground with spectroscopic grade potassium bromide (KBr) powder (2 mg of sample per 200 mg dry KBr) and then pressed into pellets with a thickness of 1 mm.

### 3.6. Ab Initio Modeling of Ligand and Ti-Complex

Based on the crystal structures of the ligand *N*,2-Dihydroxybenzamide (H_2_dihybe) and titanium(IV) complex with *N*,2-Dihydroxybenzamide (**1**), we calculated the corresponding vibrational properties. All calculations were performed with the Gaussian 09 W Revision D.01 package [67]. The initial structure of the complex used in the calculations emerged after its study by X-ray crystallography, while the structure of the H_2_dihybe was obtained from the electronic library of chemical compounds from PubChem [68]. The Density Functional Theory (DFT) using hybrid functional B3LYP, Becke’s three-parameter exchange functional with the Lee–Yang–Parr correlation functional [69,70], was chosen for all calculations. In addition, the basis set used was the 3–21 G split valence basis set. All calculations were performed without the effect of solvent, in the gaseous phase. The vibrational frequencies were calculated and scaled by a vibrational scaling factor of 0.965 to attain an acceptable agreement between the theoretical and experimental values. This is reasonable since the 3–21 G basis set used in the calculation is relatively simple and provides larger inter-atomic distances and shifted vibrational frequencies. Nevertheless, the predicted geometry for the complex resulted in reasonable parameters. Νo imaginary frequencies were observed in the results of all calculations indicating that the structures correspond to minimal points on the potential energy surface.

## 4. Conclusions

In conclusion, we synthesized a tetranuclear TOC **1** through the reaction of the hydroxamate ligand H_2_dihybe with TiCl_4_ and KOH in methyl alcohol at a pH of 1.5. The X-ray structure analysis of **1** revealed that it constitutes a rare example containing an {Ti_4_(*μ*-O)_2_} inorganic core with an almost square planar arrangement of the {Ti_4_} unit. The low molar ratio of Ti^IV^/H_2_dihybe, low pH, and room temperature lead to the formation of low nuclearity TOCs.

^1^H and ^13^C NMR solution (CD_3_OD) studies of **1** show its structural integrity in solution which is in good agreement with the high-resolution ESI-MS studies which revealed characteristic isotopic distribution envelopes attributed to the intact tetranuclear clusters containing the inorganic core {Ti_4_(*μ*-O)_2_}. The observed NMR, UV–Vis, and luminescence changes after the addition of the base to the CD_3_OD solution of **1**, are a result of acid–base equilibrium and not a change in the Ti^IV^ coordination sphere. Moreover, the decrease in the complex’s lability is due to the improved electron-donating properties of the ligand dihybe^2−^ associated with the deprotonation of its phenoxy group.

The structural features of **1** have also been investigated by means of vibrational spectroscopy revealing a *ν*(C=O) red-shift by ~10 cm^−1^, and a *ν*(N-O) blue-shift by ~20 cm^–1^ upon complexation in comparison to the free ligand H_2_dihybe. 

The solid-state spectroscopic studies of **1** revealed a band gap of 1.98 eV (band gap of TiO_2_ 3.20 eV) demonstrating not only the ability of the siderophore H_2_dihybe to stabilize rare metallic cores but to also modulate their electronic structure with potential uses in semiconducting photocatalytic applications. The origin of the dual-luminescence properties of the cluster **1** is currently under investigation.

## Data Availability

Not applicable.

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
