# Peer review of "Synthesis, Structural and Physicochemical Characterization of a Titanium(IV) Compound with the Hydroxamate Ligand N,2-Dihydroxybenzamide"

_molecules, 2021, doi:10.3390/molecules26185588_

Round 1
Reviewer 1 Report
In this work, Passadis and co-workers describe the synthesis of a new tetranuclear titanium oxo cluster stabilized by a bidentate hydroxamate ligand. The authors offer substantial characterization data, including NMR studies of solution behavior, which will be very useful for those looking to use the compound further. The work is significant for inorganic chemists engaged in the rationale design of Ti oxo cluster for applications, and synthetic chemists interested energy, and catalysis applications of such compounds. Overall, I consider the work is appropriate for Molecules, and I recommend its publication after the authors address the minor revisions, and questions noted below:
1) In the abstract, and introduction, the authors state a KOH solution with an pH = 1.5. How is this possible?
2) Keywords can be improved to reflect the findings of the paper instead of the techniques used. For example, titanium oxo cluster is intriguingly missing.
3) Introduction can have its presentation greatly improved. For example, Schemes 1 and 2 could be combined to give a better overview of hydroxamate-Ti coordination chemistry. Further, more details into the previous Ti oxo cluster reported using the same ligand should be provided, and the potential effects of temperature could be discussed.
4) Regarding previous use of H2dihybe ligands with Ti, authors should correct their statement that previous reactions reported on reference 46 were conducted in basic medium since they were carried out in the presence of carboxylic acids. This is in sharp contrast to their reaction conducted in the presence of a hydroxide base, which by definition is not a low pH. To rationalize such differences, and to strength their research question, I suggest authors to discuss the pKa of ligand species involved, as to determine the nature of species in solution. A Scheme to illustrate these equilibria and reactivity would greatly enhance the clarity and the quality of introduction.
5) Results and Discussion section would be more clear if started with a brief statement of the reaction conducted, preferably with a reaction scheme.
6) The term “envelope” to refer the isotopic distribution is unusual. Isn’t “pattern” a better term?
7) IR spectroscopy section could be more clear. Please consider the following suggestions:
7.1) Regarding the first sentence: “Consistent experimental band assignments based on comparison to theoretical spectra predicted by ab initio DFT/B3LYP calculations with Gaussian 3-21G basis sets.” It is not clear to what the “experimental band assignment” are consistent with.
7.2) Regarding the use of calculated spectra, several mismatches between predicted and experimental spectra are clearly visible in Fig 5a,b, though authors consider all bands are present in both cases. In this regard, it reads quite speculative, and I would suggest authors to remove it or move to the Supporting Information (SI). A thorough interpretation of the IR spectra, and comments on the difference between ligand and complexes spectra (Fig 5a) as done in the first paragraph of this section seem to be enough to support structural assignment.
7.3) Fig 5 could be more clear. Please consider stacking spectra instead of overlaying them. Moreover, labelling the peaks discussed in the text would greatly enhance clarity, like the authors did in Fig 6. As suggested in 7.2, removal of 5b,c is suggested.
8) NMR spectroscopy section presentation could be more clear. Please consider the following suggestions:
8.1) Move compound 2 description from Table 3, to a separate table in the SI.
8.2) Resolution of Fig 6 is not good. Please consider stacking the spectra instead of overlaying them, using boxes to highlight regions mentioned in the text, in place or in addition to the labels already used.
8.3) The spectra of the ligand in the presence of a base only supports the NMR assignments. Thus, moving it to the SI is suggested. This would include removing spectrum C from Fig 6, and including it in the SI along with Table 4 and the text related to it.
9) A final comment could be included in the section Results and Discussion to discuss reasons of obtaining a Ti4 cluster instead of cluster of higher nuclearity as reported previously. Although these reasons are not completely elucidated in the literature, a comparison between previous and current reactions conditions pointing out potential factors leading to the formation of Ti4 cluster would be of great service for those interested in the synthesis and application of these compounds.
10) Section ‘Materials and Methods’ would be better placed after Introduction of after Conclusions.
11) Considering stability of 1 in solution, the UV-Vis spectra in solution could be included as well. This would give the authors a chance to discuss also eventual changes in the absorbance when 1 is treated with a base. There is certainly an interest in compounds for which the luminescence can be tuned with the pH, and it would be interesting to see whether 1 could be considered for further applications in this area as well.
Author Response
1) In the abstract, and introduction, the authors state a KOH solution with and pH = 1.5. How is this possible?
The following sentence has been added in the text, which explains the low pH of the reaction system.
The synthesis of the TOC 1 takes place according to the following equation and the produced HCl is responsible for the low pH of the system and presumably for the mono-deprotonation of the eight ligands of 1.
4TiCl4 + 8H2dihybe + 8KOH --> [Ti4(μ-Ο)2(Hdihybe)8]Cl4 +8KCl + 4HCl + 6H2O
2) Keywords can be improved to reflect the findings of the paper instead of the techniques used. For example, titanium oxo cluster is intriguingly missing.
The reviewer is quite right, and the new most of the keywords reflect the findings of this study.
3) Introduction can have its presentation greatly improved. For example, Schemes 1 and 2 could be combined to give a better overview of hydroxamate-Ti coordination chemistry. Further, more details into the previous Ti oxo cluster reported using the same ligand should be provided, and the potential effects of temperature could be discussed.
The reviewer is quite right and first Schemes 1 and 2 were combined into Scheme 1 and second the three reported titanium/ H2dihybe oxo-titanium(IV) clusters were included along with the potential effects of temperature in the formation of various TOCs were included in the text.
4) Regarding previous use of H2dihybe ligands with Ti, authors should correct their statement that previous reactions reported on reference 47 were conducted in basic medium since they were carried out in the presence of carboxylic acids. This is in sharp contrast to their reaction conducted in the presence of a hydroxide base, which by definition is not a low pH. To rationalize such differences, and to strength their research question, I suggest authors to discuss the pKa of ligand species involved, to determine the nature of species in solution. A Scheme to illustrate these equilibria and reactivity would greatly enhance the clarity and the quality of introduction.
All the points raised by the reviewer were agreed upon and included in the text.
5) Results and Discussion section would be more clear if started with a brief statement of the reaction conducted, preferably with a reaction scheme.
The reviewer is quite right and in the revised manuscript (Results and Discussion Section) has been added the paragraph entitled ‘Synthesis of 1 and comparison with the reported higher nuclearity TOCs/H2dihybe’
6) The term “envelope” to refer the isotopic distribution is unusual. Isn’t “pattern” a better term?
Agreed and corrected.
7) IR spectroscopy section could be more clear. Please consider the following suggestions:
7.1) Regarding the first sentence: “Consistent experimental band assignments based on comparison to theoretical spectra predicted by ab initio DFT/B3LYP calculations with Gaussian 3-21G basis sets.” It is not clear to what the “experimental band assignment” are consistent with.
We agree with the reviewer’s comment and the manuscript was revised in several points. In detail, the first sentence was revised to avoid misunderstanding (page 12, section 2.3. IR Spectroscopy).
7.2) Regarding the use of calculated spectra, several mismatches between predicted and experimental spectra are clearly visible in Fig 5a,b, though authors consider all bands are present in both cases. In this regard, it reads quite speculative, and I would suggest authors to remove it or move to the Supporting Information (SI). A thorough interpretation of the IR spectra, and comments on the difference between ligand and complexes spectra (Fig 5a) as done in the first paragraph of this section seem to be enough to support structural assignment.
The Figure 5 was revised (removal of Fig. 5b and 5c) according to reviewer’s suggestions and it was added in the Supporting Information (SI) the comparison between the experimental and theoretically predicted IR absorbance spectra of the H2dihybe and Ti-complex 1 (Page 14, revised Fig. 5).
7.3) Fig 5 could be more clear. Please consider stacking spectra instead of overlaying them. Moreover, labelling the peaks discussed in the text would greatly enhance clarity, like the authors did in Fig 6. As suggested in 7.2, removal of 5b,c is suggested.
The Figure 5 was revised by stacking the spectra instead of overlaying them and added the necessary labeling according to reviewer’s suggestions (Page 14, revised Fig. 5).
8) NMR spectroscopy section presentation could be more clear. Please consider the following suggestions:
8.1) Move compound 2 description from Table 3, to a separate table in the SI.
Agreed and corrected.
8.2) Resolution of Fig 6 is not good. Please consider stacking the spectra instead of overlaying them, using boxes to highlight regions mentioned in the text, in place or in addition to the labels already used.
Agreed and corrected.
8.3) The spectra of the ligand in the presence of a base only supports the NMR assignments. Thus, moving it to the SI is suggested. This would include removing spectrum C from Fig 6, and including it in the SI along with Table 4 and the text related to it.
Agreed and corrected.
9) A final comment could be included in the section Results and Discussion to discuss reasons of obtaining a Ti4 cluster instead of cluster of higher nuclearity as reported previously. Although these reasons are not completely elucidated in the literature, a comparison between previous and current reactions conditions pointing out potential factors leading to the formation of Ti4 cluster would be of great service for those interested in the synthesis and application of these compounds.
The reviewer is quite right, and a section has been included in the Results and Discussion trying to explain why in our case a Ti4 cluster was formed in marked contrast to the reported higher nuclearity clusters with the same ligand.
10) Section ‘Materials and Methods’ would be better placed after Introduction of after Conclusions.
Agreed and corrected.
11) Considering stability of 1 in solution, the UV-Vis spectra in solution could be included as well. This would give the authors a chance to discuss also eventual changes in the absorbance when 1 is treated with a base. There is certainly an interest in compounds for which the luminescence can be tuned with the pH, and it would be interesting to see whether 1 could be considered for further applications in this area as well.
The solution UV-vis and luminescence spectra of cluster 1 without and with the addition of base were included in the text.
Reviewer 2 Report
The authors report on an interesting titanium oxo cluster and perform extensive characterisation of this new crystal structure contrasting it with the original ligand.
Aside from a few aspects detailed below which need correction and elaboration I believe this work is based on sound evidence, of scholarly presentation and will be of interest to the readership of Molecules and will gladly recommend it for publication.
I would recommend a further addition to the introduction since this paper deals with a Ti4 core but instead a notable mu4-oxo ligand : https://doi.org/10.3390/ma11091661
My biggest disappointment is with the computational part and the IR spectral treatment. The 3-21G Pople basis set is by now of historical interest and of very little use save for educational purposes. Even still without any polarisation functions, it is not surprising that the calculated IR spectrum has a poor agreement with the experimental one. There are plenty of missing peaks, wrong intensities etc.
With regards to the discussion there is one sentence which might be contentious:
“An intense band at ~800 cm–1 complex which is absent in the spectrum of the H2dihybe”
This is not strictly true, upon examining Figure 5b) and 5c) it could well be that the tiny peak at around 750 has simply slightly stiffened its vibrational modes upon coordination. The calculation itself would reveal which vibrational mode this pertains to and the authors could draw up a figure or a table stating which atoms are moving/rocking in this frequency range.
My recommendation would be to use a better basis set (at least of DZP quality) for the calculation and to better explore these discrepancies of the ligand/complex spectra by looking at the most important vibrational modes. A pictorial representation of the atomic displacements in these fingerprint vibrational modes would be ideal and instructive to the reader. The complex has D2 point group symmetry which I assume was taken advantage of in the calculations (incidentally this should be stated in the computational details) so a recalculation should not be so time consuming.
The band gap value of 1.98 eV was mentioned in the conclusions. How does it compare with the ubiquitous TiO2 value?
Some minor spelling errors:
P2. “basic pH environment where the ligand [was] found to interact”
P17 “3-21G split val[e]nce basis”
Author Response
1) I would recommend a further addition to the introduction since this paper deals with a Ti4 core but instead a notable mu4-oxo ligand : https://doi.org/10.3390/ma11091661
The reviewer is quite right, and this reference was included in the text (ref 11)
2) My biggest disappointment is with the computational part and the IR spectral treatment. The 3-21G Pople basis set is by now of historical interest and of very little use save for educational purposes. Even still without any polarisation functions, it is not surprising that the calculated IR spectrum has a poor agreement with the experimental one. There are plenty of missing peaks, wrong intensities etc.
With regards to the discussion there is one sentence which might be contentious:
“An intense band at ~800 cm–1 complex which is absent in the spectrum of the H2dihybe”
This is not strictly true, upon examining Figure 5b) and 5c) it could well be that the tiny peak at around 750 has simply slightly stiffened its vibrational modes upon coordination. The calculation itself would reveal which vibrational mode this pertains to and the authors could draw up a figure or a table stating which atoms are moving/rocking in this frequency range.
My recommendation would be to use a better basis set (at least of DZP quality) for the calculation and to better explore these discrepancies of the ligand/complex spectra by looking at the most important vibrational modes. A pictorial representation of the atomic displacements in these fingerprint vibrational modes would be ideal and instructive to the reader. The complex has D2 point group symmetry which I assume was taken advantage of in the calculations (incidentally this should be stated in the computational details) so a recalculation should not be so time consuming.
We revised this part to avoid misunderstanding. Indeed, the intense band at ~800 cm–1 is observed only in the spectrum of the complex, while is absent in the spectrum of the H2dihybe and is attributed to the titanium complex formation and more specifically is assigned to the high-energy ν(Ti2–μ-O) stretching modes. Please note that the two bands observed in lower frequencies in the spectrum of the H2dihybe are also present in the spectrum of the complex. More specifically, the low-frequency band of H2dihybe observed at ~780 is red-shifted to 770 cm-1 in the spectrum of complex, while the band of H2dihybe at ~744 cm-1 remains in the same frequency with much lower absorbance. We revised completely Fig. 5. We added in Supporting Information the comparison between the experimental and theoretically predicted IR absorbance spectra of the H2dihybe and Ti-complex 1.
All theoretical spectra have been estimated by ab initio DFT/B3LYP calculations with the Los Alamos National Laboratory 2 double zeta (LanL2DZ) split-valence basis set according to reviewer’s suggestions (double zeta quality). The specific basis set is an ideal choice for quantum mechanical calculations for complexes whose centers are first-row transition metals, such as titanium. The calculations of such molecules with LANL2DZ basis set and the fact that the complex has D2 point group symmetry are characterized by a relatively short computational time in agreement with reviewer’s suggestions. Considering the LANL2DZ basis set, a reason for the reduction in the computational time is that ECP (Effective Core Potential) plus double zeta on Na-Bi is used in this basis set. ECP describes the inner electron orbitals and so no basis functions are required for them. Another advantage of this basis set is that include relativistic effects, but it does not include polarization functions. These details have been added in the revised manuscript (page 12, section 2.3. IR Spectroscopy).
Furthermore, we added the necessary labeling for the bands discussed in the text by providing the atoms involved in these fingerprint vibrational modes for the reader’s convenient (Page 14, revised Fig. 5).
3) The band gap value of 1.98 eV was mentioned in the conclusions. How does it compare with the ubiquitous TiO2 value?
Agreed and corrected.
Some minor spelling errors:
P2. “basic pH environment where the ligand [was] found to interact”
Agreed and corrected.
P17 “3-21G split val[e]nce basis”
Agreed and corrected.
Reviewer 3 Report
The paper by T. A. Kabanos et al gives a complete and very thorough characterization of a single titanium cluster compound. With the results of the analyses, no doubt is left about its true identity. Because of the quality of the work and the presentation I suggest publication of this paper in Molecules. I do have two minor comments.
- A deprotonation of the ligand at pH=1.5, although not complete but still, is surprising.
- A couple of sentences about the hydroxamate binding modes in other coordination compounds would be welcome.
Author Response
- A deprotonation of the ligand at pH=1.5, although not complete but still, is surprising.
The ligation of the ligand to TiIV makes the hydroxamic proton [~C(O)NHO-H] more acidic, and thus, despite the low pH results in the deprotonation of the ligand.
- A couple of sentences about the hydroxamate binding modes in other coordination compounds would be welcome.
Figure S1 has been added in the supporting information with the main binding modes of metal-hydroxamates.